

# Multifactorial analysis of temperature, solute-to-solvent ratio, and ultrasound amplitude on the extraction of phenolic and antioxidant compounds from *Aloysia citriodora* Palau leaves

Osvaldo Alvarez Cortes[1,*], Héctor Eduardo Martinez Flores[1,*], Martha-Estrella García-Pérez[1], Juan Carlos González-Hernández[2], Martha Eva Viveros-Sandoval[3], Mariano Martinez-Vazquez[4], Noemi Silva Jimenez[5] and María Carmen Bartolomé-Camacho[1]

[1] Facultad de Químico Farmacobiología, Universidad Michoacana de San Nicolás de Hidalgo, Morelia, Michoacán, Mexico

[2] Tecnológico Nacional de México/ Instituto Tecnológico de Morelia, Morelia, Michoacán, México

[3] Facultad de Ciencias Médicas y Biológicas ''Dr Ignacio Chávez, Universidad Michoacana de San Nicolás de Hidalgo, Morelia, Michoacán, Mexico

[4] Instituto de Química, Universidad Nacional Autónoma de México, Ciudad de México, CDMX, Mexico

[5] Instituto de Investigaciones Biomédicas, Universidad Nacional Autónoma de México, Ciudad de México, CDMX, Mexico

[*] These authors contributed equally to this work.

Corresponding author
Héctor Eduardo Martinez Flores,
hector.martinez.flores@umich.mx

## ABSTRACT

Ultrasonic-assisted extraction was applied to *Aloysia citriodora* Palau leaves. To achieve this, a set of extraction parameters—temperature, amplitude, and solute-to-solvent ratio—was evaluated. A $3^3$ factorial design was implemented, generating 27 treatments, from which total phenolic content, total flavonoid content, and antioxidant capacity against 2,2′-azino-bis(3-ethylbenzothiazoline-6-sulfonic acid) (ABTS$^{•+}$), 2,2-diphenyl-1-picrylhydrazyl (DPPH$^{•}$), and hydroxyl (OH$^{•}$) radicals were assessed. The extract showing the highest free radical scavenging capacity was obtained at 75 °C, a solute-to-solvent ratio of 1:15, and a wave amplitude of 25%. This extract contained 87.74 mg gallic acid equivalent (GAE)/g extract and 76.82 ± 6.21 mg quercetin equivalent (QE)/g extract. Antioxidant capacity was expressed as the half-maximal inhibitory concentration (IC$_{50}$, μg/mL), with values of 9.08 ± 0.08 μg/mL for ABTS$^{•+}$, 35.58 ± 0.50 μg/mL for DPPH$^{•}$, and 0.72 ± 0.007 mg/mL for OH$^{•}$. The *A. citriodora* leaf extract demonstrated strong potential for use in the development of new products containing bioactive molecules capable of preventing degenerative diseases.

## INTRODUCTION

Polyphenolic compounds derived from medicinal plants have attracted considerable attention due to their potential role as precursors or fundamental components in the synthesis of various pharmaceutical agents. As *Wangchuk (2018)* noted, 73% of existing pharmaceutical products/medicines are derived from natural products, including medicinal compounds from traditional medicines. The technologies currently employed for the extraction of these compounds include supercritical fluid extraction, a process that yields clean extracts free of residual solvents. Microwave-assisted extraction is another method that has been shown to be rapid and efficient in the extraction of secondary metabolites. However, it should be noted that the heating process may result in the degradation of certain compounds, such as glucosides, terpenes, and flavonoids, due to the denaturation of these volatile compounds (*Herrero et al., 2010*). In recent years, novel processes, including ultrasound technology, have emerged as effective methodologies for bioactive compound extraction. Ultrasound is a technology that has been used as an efficient alternative for secondary metabolite extraction from plants. This technology has been demonstrated to yield higher results, reduce extraction times, and decrease operating costs in comparison with conventional methods, including maceration and Soxhlet extraction (*Jerman & Mozetič, 2011*; *Osorio-Tobón, 2020*). *Topete-Betancourt et al. (2023)* observed a greater efficiency in the extraction of phytocannabinoids from Cannabis using ultrasonic-assisted extraction. Solid–liquid extraction, the most widely implemented process, involves the separation of one or more substances (solutes) contained within a solid matrix (carrier phase) using solvents (*Priego-Capote, 2021*). The extraction rate of the molecules of interest is influenced by several factors, including the size of the particles, the type of solvent, and the temperature. These parameters have been modified to achieve the optimal yield and prevent bioactive compound degradation in plants, which has generated interest in the pharmaceutical and food industries (*Athanasiadis et al., 2024*). Furthermore, the optimization of variables such as the solute/solvent ratio, ultrasound amplitude, and temperature is essential to ensure the maximal efficiency of the phenolic compound extraction process. These conditions directly influence the disruption of the plant matrix, the solubility of the compounds, and the prevention of their thermal degradation (*Hashemi et al., 2022*). The adjustment of these parameters has been shown to yield a number of benefits, including improvements in the yield and quality of the extract. Additionally, these adjustments provide significant environmental benefits, such as a reduction in solvent and energy use, as well as economic benefits, including shorter extraction times. Finally, the preservation of bioactivity in the compounds during extraction has been demonstrated to be a functional health benefits. This optimization is of particular relevance for applications in the pharmaceutical, food, and cosmetic industries, where sustainable and highly efficient processes are imperative to obtain natural bioactive ingredients.

The family Verbenaceae comprises approximately 2,000 species and over 100 genera, distributed across a broad spectrum of geographical regions, encompassing tropical, subtropical, and temperate zones on a global scale (*Polumackanycz et al., 2022*). *Aloysia citriodora* Palau, a member of the Verbenaceae family commonly referred to as lemon

verbena, is extensively cultivated in South America and distributed globally as an ornamental plant. The plant properties have been demonstrated to include a range of benefits, including its capacity to relax the nervous system, act as an anti-inflammatory agent, provide analgesic effects, function as an anti-spasmodic agent, and exhibit antibacterial properties (*Bahramsoltani et al., 2018*). *Aloysia citriodora* leaves are abundant in phenolic compounds, the primary component of which is verbascoside, which has anti-inflammatory activity. The chemical composition of the plant has been identified as containing luteolin 7-O-diglucuronide, p-coumaric acid, phenylethanoid, eucovoside, and martynoside (*Pereira et al., 2017*; *Li et al., 2018*; *Sun et al., 2020*). Extracts of *A. citriodora*, containing phenolic compounds, have been demonstrated to possess antibacterial, anticoagulant, antioxidant, and cytoprotective effects. As stated by *Bahramsoltani et al. (2018)*, the substance exhibits antinociceptive and anti-inflammatory effects, as well as cardiovascular disease. Traditionally, plants with high concentrations of verbascoside have been used in folk medicine to treat inflammation disorders (*Georgiev et al., 2012*).

The objective of the present study was to obtain an extract of phenolic compounds from *A. citriodora* leaves by evaluating different ultrasound processing conditions. Furthermore, an evaluation was conducted to ascertain the antioxidant activity of *A. citriodora* leaves. This evaluation involved the utilization of three distinct methods: the first method entailed the utilization of 2,2′-azino-bis (3-ethylbenzothiazoline-6-sulfonic acid) (ABTS$^{\bullet+}$) scavenging, the second method involved the utilization of 2,2-diphenyl-1-picrylhydrazyl (DPPH$^{\bullet}$) scavenging, and the third method involved the utilization of hydroxyl radicals.

## MATERIALS AND METHODS

### Collection of *A. citriodora* leaves

The leaves were collected from the cedrón plant (*Aloysia citriodora*) in August 2022, during the summer season, at the following coordinates: latitude 19.6912985, longitude −101.2054516, and an altitude of 1,909 m, in Morelia, Michoacán, Mexico. The age of the plant is estimated to be approximately 25 years. The specimen was taxonomically identified using the database of the Herbarium of the Universidad Michoacana de San Nicolás de Hidalgo. Taxonomic identification was based on a comprehensive analysis of the leaves, flowers, and fruits. The leaves were subjected to shade-drying at room temperature for a period of 15 days. Prior to the drying process, a meticulous examination of the plant material was conducted to guarantee its freedom from pests or indications of physiological stress. Subsequently, the samples were ground and sieved using a 0.3331 inch mesh.

### Phenolic extracts: full factorial design

The conditions used to extract phenolic compounds in this study were based on previous studies that evaluated various ultrasound-assisted extraction parameters. According to these studies, the temperature range was from 30 to 90 °C, the solute-to-solvent ratio was from 1:5 to 1:100, the extraction time was from 10 min to 48 h, and the ultrasound amplitude was from 30% to 50% (*Tena-Rojas et al., 2022*; *Tranquilino-Rodríguez & Martínez-Flores, 2023*).

The extracts were obtained using a VCX 500 ultrasound-assisted extraction system (Sonics & Materials, Inc.) operating at 20 kHz with a 220-B type probe. A full factorial design was implemented to identify optimal conditions for the ultrasound process (Table 1). This design incorporated the following variables—ratio of dried *A. citriodora* leaves to solvent: 1:5, 1:10, and 1:15 (where 1:5 equals 1 g in 5 mL), temperature: 30 °C, 50 °C, and 75 °C and amplitude: 20%, 25%, and 35%. The extraction time was set at 10 min based on preliminary tests and in data found in the literature. For example, studies on *Carica papaya* Linn leaves reported that the optimal extraction time was 9 min (*Puramshetti et al., 2025*). *Zerihun-Chala et al. (2024)* also found that 10 min was optimal for extracting phenolic compounds from *A. remota* Benth leaves. Furthermore, numerous reports have demonstrated that temperatures exceeding 85 °C and extended exposure periods can result in the degradation of target compounds. This study evaluated solute/solvent ratios of 1:5, 1:10, and 1:15 (g/mL) for polyphenolic compound extraction to determine the most efficient ratio in terms of yield and phenolic content. This approach is supported by previous studies in this field. For instance, *Rahmawati et al. (2024)* reported that a ratio of 1:8 (g/mL) was optimal for the microwave-assisted extraction of polyphenols from banana peels, with concentrations reaching up to 354.02 mg GAE/g.

Additionally, *Jeganathan et al. (2014)* identified an optimal ratio of 1:8.7 g/mL for red grapes, which had a substantial impact on anthocyanins and flavonoids. Similarly, *Shabri & Rohdiana (2016)* found that a 1:15 (w/v) ratio produced the best polyphenol extraction from green tea with 70% acetone. The collective findings of these studies validate the selected range (1:5–1:15) as appropriate for evaluating extraction efficiency.

The amplitude percentage was determined based on the operational limits of the equipment, ensuring that it was not lower than 20% nor higher than 40%. The experimental workflow diagram is provided in the Material S4. The extracts were filtered using Whatman No. 1 paper and stored at −20 °C in amber glass containers. This method was employed to minimize light exposure and prevent oxidation. All extractions were carried out under controlled temperature conditions. The extracts were immediately cooled in an ice bath after extraction to prevent thermal degradation of phenolic compounds. The following response variables were employed to determine the most effective combination: quantification of total phenolic and flavonoid content and antioxidant activity of the extracts against $ABTS^{•+}$ (2,2′-azino-bis(3-ethylbenzothiazoline-6-sulfonic acid)), $DPPH^{•}$ (2,2-diphenyl-1-picrylhydrazyl), and hydroxyl radicals ($OH^{•}$).

## Determination of phenolic compounds and antioxidant tests
### Total phenolic and total flavonoid contents
The Folin–Ciocalteu (FC) assay is based on the reaction of phenolic compounds with the FC reagent, which results in a blue color. This color is measured spectrophotometrically at 750 nm and is expressed as gallic acid equivalents. The FC reagent was prepared by diluting it 1:1 in distilled water. A 2% sodium carbonate solution was prepared in distilled water and sonicated for 2 min at room temperature. A 0.2 mg/mL stock solution of gallic acid in distilled water was used to construct the calibration curve (*Taga, Miller & Pratt, 1984*). The flavonoid content was measured using the method described by
**Table 1 Total phenolic content, total flavonoid content and radical scavenging activity of *Aloysia citrodora* leaf extracts obtained using the ultrasound process.**

| Treatment | Independent variables | | | | | Response variables | | |
|---|---|---|---|---|---|---|---|---|
| | T (°C) | R | A | Total phenolics (mg GAE/g) | Total flavonoids (mg QE/g) | OH$^\bullet$ IC50 (mg/mL) | DPPH$^\bullet$ IC50 ($\mu$g/mL) | ABTS$^{\bullet+}$ IC50 ($\mu$g/mL) |
| T3R2A2 | 75 | 1:10 | 25 | 133 ± 3.1[A] | 86 ± 1.6[BCD] | 0.75 ± 0.01[MN] | 480 ± 0.4[FGH] | 10.3 ± 0.01[NO] |
| T3R3A1 | 75 | 1:15 | 20 | 93 ± 8.8[B] | 94 ± 8.3[AB] | 0.86 ± 0.01[LMN] | 43 ± 0.6[FGH] | 10.6 ± 0.01[MNO] |
| T3R3A2 | 75 | 1:15 | 25 | 88 ± 17.6[B] | 77 ± 6.2[E] | 0.72 ± 0.01[N] | 35 ± 0.5[FGH] | 9.1 ± 0.08[O] |
| T2R3A3 | 50 | 1:15 | 35 | 80 ± 13.9[BC] | 99 ± 3.1[A] | 1.21 ± 0.01[KLM] | 97 ± 1.9[EFGH] | 15.1 ± 0.18[L] |
| T3R1A3 | 75 | 1:5 | 35 | 69 ± 4.0[CD] | 58 ± 0.7[F] | 3.19 ± 0.06[C] | 162 ± 3.2[CDEFGH] | 37.3 ± 0.15[H] |
| T2R3A1 | 50 | 1:15 | 20 | 68 ± 3.4[CD] | 93 ± 2.0[ABC] | 0.96 ± 0.01[LMN] | 106 ± 2.1[EFGH] | 13.1 ± 0.51[LMN] |
| T2R2A1 | 50 | 1:10 | 20 | 67 ± 4.7[CD] | 84 ± 3.5[CDE] | 1.97 ± 0.06[FGHI] | 117 ± 5.1[DEFGH] | 11.5 ± 0.51[MNO] |
| T3R3A3 | 75 | 1:15 | 35 | 67 ± 7.2[CD] | 94 ± 2.8[AB] | 1.29 ± 0.01[JKL] | 650 ± 0.9[EFGH] | 13.7 ± 0.05[LM] |
| T2R2A2 | 50 | 1:10 | 25 | 66 ± 4.1[CD] | 9 ± 6.1[JK] | 0.88 ± 0.01[LMN] | 124 ± 12.0[DEFGH] | 14.9 ± 0.18[L] |
| T2R1A3 | 50 | 1:5 | 35 | 61 ± 1.4[DE] | 91 ± 4.3[ABCD] | 2.04 ± 0.62[FGH] | 115 ± 3.3[EFGH] | 37.2 ± 1.49[H] |
| T2R1A2 | 50 | 1:5 | 25 | 59 ± 2.8[DEF] | 83 ± 1.3[DE] | 1.89 ± 0.03[GHI] | 103 ± 3.4[EFGH] | 29.8 ± 0.40[I] |
| T2R1A1 | 50 | 1:5 | 20 | 56 ± 1.8[DEFG] | 60 ± 1.3[F] | 2.38 ± 0.06[DEF] | 124 ± 2.4[DEFGH] | 40.6 ± 0.15[EFG] |
| T1R2A3 | 30 | 1:10 | 35 | 56 ± 3.9[DEFG] | 64 ± 1.3[F] | 0.90 ± 0.01[LMN] | 97 ± 11.0[EFGH] | 10.9 ± 0.16[MNO] |
| T1R2A2 | 30 | 1:10 | 25 | 53 ± 0.7[DEFGH] | 37 ± 2.3[GH] | 1.98 ± 0.02[FGHI] | 108 ± 1.1[EFGH] | 28.8 ± 0.67[IJ] |
| T3R2A3 | 75 | 1:10 | 35 | 49 ± 1.9[EFGH] | 62 ± 2.5[F] | 1.69 ± 0.01[HIJ] | 99 ± 3.6[EFGH] | 18.9 ± 0.18[K] |
| T2R3A2 | 50 | 1:15 | 25 | 45 ± 2.3[EFGH] | 76 ± 2.6[E] | 1.23 ± 0.01[JKL] | 96 ± 4.1[EFGH] | 14.6 ± 0.02[L] |
| T1R1A3 | 30 | 1:5 | 35 | 45 ± 0.9[EFGH] | 13 ± 1.1[J] | 1.54 ± 0.08[IJK] | 198 ± 0.6[CDEF] | 31.8 ± 0.09[I] |
| T1R1A1 | 30 | 1:5 | 20 | 43 ± 4.4[FGH] | 32 ± 1.6[HI] | 2.17 ± 0.08[EFG] | 310 ± 6.1[BC] | 41.0 ± 0.84[DEF] |
| T3R2A1 | 75 | 1:10 | 20 | 41 ± 0.5[GH] | 26 ± 1.6[I] | 2.66 ± 0.02[D] | 175 ± 1.5[CDEFGH] | 38.4 ± 0.64[FGH] |
| T3R1A1 | 75 | 1:5 | 20 | 40 ± 2.0[GH] | 25 ± 0.1[I] | 3.46 ± 0.01[BC] | 182 ± 0.5[CDEFG] | 44.2 ± 0.60[CD] |
| T2R2A3 | 50 | 1:10 | 35 | 40 ± 2.9[GH] | 44 ± 1.8[G] | 2.31 ± 0.02[DEFG] | 293 ± 23.5[BCD] | 26.4 ± 0.61[J] |
| T3R1A2 | 75 | 1:5 | 25 | 40 ± 2.4[GH] | 24 ± 2.5[I] | 3.22 ± 0.28[C] | 161 ± 5.4[CDEFGH] | 37.6 ± 0.40[GH] |
| T1R1A2 | 30 | 1:5 | 25 | 38 ± 1.9[H] | 7 ± 1.4[JK] | 1.87 ± 0.02[GHI] | 226 ± 3.7[CDE] | 46.1 ± 0.22[C] |
| T1R2A1 | 30 | 1:10 | 20 | 21 ± 0.9[I] | 11 ± 0.8[JK] | 2.58 ± 0.01[DE] | 412 ± 9.7[B] | 63.1 ± 0.27[B] |
| T1R3A2 | 30 | 1:15 | 25 | 21 ± 1.0[I] | 25 ± 0.2[I] | 2.70 ± 0.02[D] | 459 ± 61.5[B] | 37.6 ± 0.77[GH] |
| T1R3A3 | 30 | 1:15 | 35 | 15 ± 1.5[IJ] | 11 ± 1.0[JK] | 3.90 ± 0.03[B] | 232 ± 7.5[CDE] | 42.4 ± 1.63[DE] |
| T1R3A1 | 30 | 1:15 | 20 | 3 ± 2.1[J] | 8 ± 0.8[JK] | 12.23 ± 0.29[A] | 1,790 ± 291.0[A] | 81.6 ± 4.41[A] |
| Oligopin | | | | 11 ± 0.4[IJ] | 26 ± 2.1[K] | 1.04 ± 0.04[LMN] | – | – |
| Gallic acid | | | | | | – | 2.35 ± 0.09[H] | 0.91 ± 0.02[P] |
| Ascorbic acid | | | | | | – | 8.25 ± 0.18[GH] | – |
| Trolox | | | | | | – | – | 0.12 ± 0.07[P] |

**Notes.**

T, Temperature; R, Ratio leaf extract/water; A, Amplitude.

Different letters (A, B, C, D, *etc.*) in the same column indicate significant differences at $p < 0.05$ (ANOVA, followed by Tukey's test) $n = 3$.

IC50, half of the maximum effective concentration.

*Afrasiabian et al. (2018)*. A 2% (m/v) aluminum chloride ($AlCl_3$) solution was prepared by dissolving 0.2 g of $AlCl_3$ in 10 mL of 50% methanol. The solution was then subjected to ultrasonication for 5 min. A quercetin stock solution at a concentration of 0.02 mg/mL was used to construct the calibration curve. The obtained coefficients of determination were $R^2 = 0.999$ for flavonoids and $R^2 = 0.996$ for phenolic compounds.

Both curves are included in the Material S5.

### Determination of antioxidant activity

*2,2-Diphenyl-1-picrylhydrazyl (DPPH●) scavenging activity.* This method involved measuring the scavenging of free radicals by antioxidant compounds using DPPH$^\bullet$ due to the presence of antioxidant substances sensitive to spectrophotometric determination at 515 nm. Ascorbic acid at 0.02 mg/mL and gallic acid at 33.3 µg/mL in water were used as standards (*Brand, Cuvelier & Berset, 1995*; *Valencia-Avilés et al., 2018*). The DPPH reagent was prepared at a concentration of $9.13 \times 10^{-5}$ mol L$^{-1}$ using methanol as a diluent. After preparing the radical, the absorbance was measured and adjusted to 0.90. To obtain the inhibition percentage, Eq. (1) was used.

$$\% \, inhibition = \left( \frac{(A0 - A1)}{A0} \right) * 100 \tag{1}$$

where $A0$ = Absorbance of DPPH, $A1$ = Absorbance of the extract + DPPH.

*2-2′-azino bis (3-ethylbenzothiazoline-6-sulfonic acid (ABTS$^{\bullet+}$) scavenging activity.* The radical cation derived from 2,2′-azino-bis(3-ethylbenzothiazoline-6-sulfonic acid) (ABTS), known as ABTS$^{\bullet+}$, is one of the most commonly used to evaluate the antioxidant efficacy of pure compounds and complex mixtures. The ABTS$^{\bullet+}$ decolorization assay measures total antioxidant capacity in both lipophilic and hydrophilic substances. The ABTS$^{\bullet+}$ radical was prepared by dissolving a 0.05 g ABTS tablet in 13.8 mL of water and mixing it in a 1:1 ratio with 2.45 mM potassium persulfate dissolved in methanol. Radical formation occurred at 22 °C in the dark for 24 h. Then, the absorbance of the radical was measured and adjusted to 0.90. Trolox (6-hydroxy-2,5,7,8-tetramethylchroman-2-carboxylic acid), at a concentration of 70 µg/mL in methanol, and gallic acid, at a concentration of 3.5 µg/mL in water, were used as standards. To calculate the inhibition percentage, Eq. (2) was used. Trolox equivalent antioxidant capacity (TEAC) was determined according to Eq. (3).

$$\% \, inhibition = \left( \frac{(A0 - A1)}{A0} \right) * 100 \tag{2}$$

where $A0$ = Absorbance of ABTS, $A1$ = Absorbance of the extract + ABTS.

$$TEAC = \left( \frac{(IC50 \, Trolox)}{IC50 \, Sample} \right) * 100. \tag{3}$$

*Hydroxyl radical (OH$^\bullet$) scavenging activity.* The antioxidant activity of the extracts against the hydroxyl radical was determined according to the methods described by *Smirnoff & Cumbes (1989)* and *Valencia-Avilés et al. (2018)*. Different concentrations of *A. citriodora* leaf extract was added to 600 µL of dimethyl sulfoxide at various concentrations. Then, 200 µL of FeSO$_4$ (eight mM) and 167 µL of H$_2$O$_2$ (2%) were added. The reaction was initiated by adding 600 µL of a three mM salicylic acid solution, after which the mixture was incubated at 30 °C for 30 min. Absorbance was measured at a wavelength of 510 nm. The inhibition percentage was calculated using the following formula:

$$\% \, inhibition = \left( \frac{(A0 - (A2 - A1))}{A0} \right) * 100.$$

$A2$=Absorbance of the extract+hydroxyl radical:

$A0$=Absorbance of hydroxyl radical:

$A1$=Absorbance of the extract

The effective concentration ($IC_{50}$), defined as the amount of extract required to reduce the concentration of the reactive species by 50%, was calculated in all antioxidant capacity tests for hydroxyl, DPPH, and ABTS radicals. All analyses were performed in triplicate. The affinity of each radical is determined by the polarity and charge of the antioxidants. Therefore, measuring these three radicals provides important information. Hydrophilic and polar antioxidants, such as certain polyphenols and vitamin C, are more effective against polar radicals, such as hydroxyl and ABTS. In contrast, lipophilic antioxidants, such as vitamin E, interact more strongly with the nonpolar DPPH radical.

## Statistical analysis

The data were analyzed using a one-way analysis of variance (ANOVA) with JMP software (version 8.0). Tukey's multiple comparison test was then performed to determine statistical differences in the parameters. Values of $P \leq 0.05$ were considered statistically significant.

# RESULTS

## Sample identification

The specimen obtained was a deciduous shrub, measuring between two and six m in height, characterized by a rough texture and a lemon-like fragrance. The leaves are characterized by a blade measuring 30–110 × 7–25 mm, 3-verticillate trichomes, and stiff, antrorose trichomes with a diameter of 0.1–0.2 mm (see Fig. 1A). The calyx measures 2–3.5 mm, while the corolla measures 3.5–5.5 mm, with a white or purple coloration, as shown in Fig. 1B. Figure 1C illustrates an obovoid or oblong fruit, with mericarps measuring 1.3–1.8 × 0.5–0.6 mm, flat-convex, hairy at the apex, and brown in color. A comparison was made with specimens of the same species that had been deposited in the herbarium of the Universidad Michoacana de San Nicolás de Hidalgo in Morelia, Michoacán, Mexico. These specimens had folio numbers 27,450 and 23,355, respectively. The comparison was also made with digital herbarium images, such as SEINET. The specimen was formally designated as *Aloysia citriodora* Palau.

## Ultrasound-assisted extraction

The extracts were obtained using the VCX 500 Ultrasonic Processor with the extraction parameters considered in the experimental arrangement (Table 1). A $3^3$-factorial design with three factors and three levels was employed, yielding 27 extracts. The extraction time was 10 min. The nomenclature of each extract was selected according to Table, in which T denotes the temperature variable, with three levels numbered from 1 to 3, corresponding to 1 = 30 °C, 2 = 50 °C, and 3 = 75 °C. The variable leaf and water concentration is indicated by the letter R, which has three levels, numbered from 1 to 3, corresponding to 1 = 1:5, 2 = 1:10, and 3 = 1:15 (g/mL). The variable amplitude is designated by the letter A, with three levels numbered from 1 to 3, corresponding to 1 = 20%, 2 = 25%, and 3 =

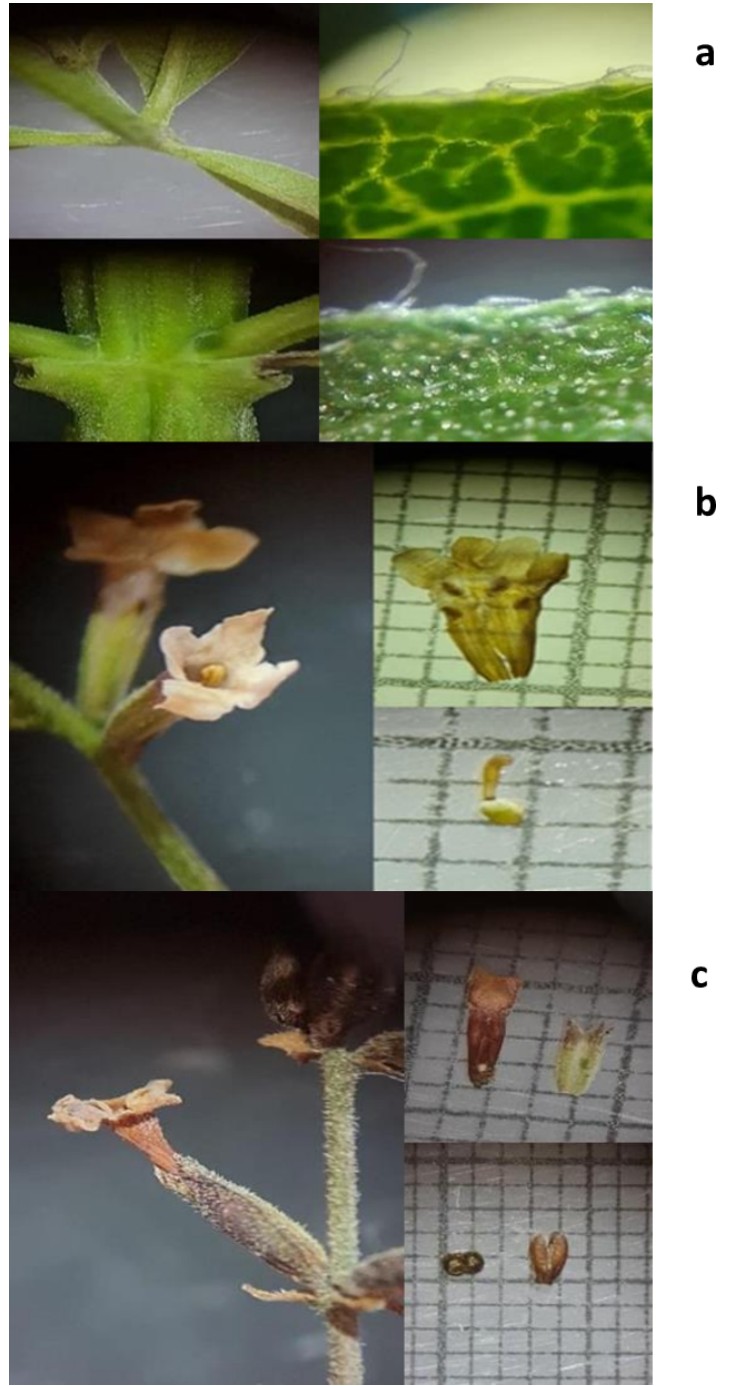

**Figure 1** (A) Leaves, (B) flowers, (C) fruits of *Aloysia citriodora* Palau.

35%. For instance, T1R1A1 corresponds to a temperature of 30 °C, a solute/solvent ratio of 1:5, and an amplitude of 20%.

## Total phenolic and total flavonoid contents

The total phenolic content is shown in Table 1 and expressed in mg equivalents of gallic acid (mg GAE/g of extract). The sample with the highest amount was T3R2A2, which had 132.84 ± 3.12 mg GAE/g. In a related study, *Wernert et al. (2009)* obtained 51.85 mg of tannic acid equivalents/g dry material from aqueous extracts of *A. citriodora*. *Al-Qaraleh & Tarawneh (2016)* evaluated methanolic extracts of *A. triphylla* plants and yielded 62.16 mg GAE/g of plant extract. In the present study, the pine bark extract (Oligopin®) exhibited a concentration of 10.90 ± 0.41 mg GAE/g. As indicated by the results of the study, multiple treatments exhibited a higher level of total phenolic compounds in comparison to the Oligopin sample.

The total flavonoid content is presented in Table 1. The sample with the highest amount was T2R3A3, which had 98.96 ± 31.13 mg QE/g; this sample yielded a higher amount of total flavonoid compounds compared to the 7.01 mg quercetin equivalents/g dry leaves of aqueous extracts of *A. citriodora* observed by another author (*Moeina, Zarshenasc & Etemadfarda, 2014*). When comparing the results obtained by *Athanasiadis et al. (2024)*, who used ultrasound-assisted extraction in *A. citriodora* L. and reported values of 40.66–150.96 mg GAE/g with exposure times ranging from 20 to 150 min, similar values to those in the present study are observed, although with a shorter exposure time.

## DPPH and ABTS scavenging activity

Gallic acid and ascorbic acid were utilized as positive controls, yielding optimal performance in terms of inhibition and concentration (see Table 1). The values obtained were 2.35 ± 0.09 µg/mL and 8.25 ± 0.18 µg/mL, respectively, indicating 50% inhibition of the DPPH$^\bullet$ radical. The aqueous extracts that exhibited a lower concentration for radical inhibition were T3R3A2, T3R2A2, and T3R3A1, with values of 35.58 ± 0.50, 47.90 ± 0.39, and 42.64 ± 0.58 µg/mL, respectively, to inhibit 50% of the DPPH$^\bullet$ radical (Table 1). The relationship between temperature and solute-to-solvent ratio is significant in the context of molecular extraction, particularly concerning its capacity to inhibit DPPH radical activity. In their study, *Cheurfa & Allem (2015)* evaluated the hydroalcoholic extract of *A. citriodora*, finding significantly higher total phenolic and flavonoid contents compared to the aqueous extract. However, the aqueous extract demonstrated a higher $IC_{50}$ value (27.40 ± 0.1 mg/mL) for scavenging the DPPH$^\bullet$ radical, indicating a potential difference in antioxidant activity between the two extracts. In contrast, *Al-Qaraleh & Tarawneh (2016)* obtained an $IC_{50}$ concentration of 0.8 mg/mL in *A. triphylla* leaf extracts, requiring a higher concentration to eliminate 50% of the radicals compared to the extracts obtained in this study using ultrasound-assisted extraction, which required a lower concentration of 35.58 µg/mL to achieve the same 50% radical elimination.

An ABTS$^{\bullet+}$ radical inhibition curve was plotted using gallic acid and trolox as standards, yielding 50% inhibition of the ABTS radical at concentrations of 0.91 ± 0.02 and 0.12 ± 0.07 µg/mL, respectively (see Table 1 for details). Treatment T3R3A2 exhibited a

lower concentration, with an inhibition rate of $9.1 \pm 0.08$ µg/mL, corresponding to 50% inhibition of the ABTS$^{\bullet+}$ radical. A study by *Hernández et al. (2015)* examined the antioxidant properties of 20 plants utilized in Mexican traditional medicine. The study reported that the plants gobernadora, "hierba dulce", and "nogal" exhibited the highest ABTS$^{\bullet+}$ radical scavenging capacity, with IC$_{50}$ values of 1.67, 2.68, and 2.70 µg/mL, respectively. Conversely, higher concentrations were required to achieve IC$_{50}$ values of 6.90 and 11.18 µg/mL in rosemary and chamomile plants, respectively. The trolox equivalent activity (TEAC) of the "T3R3A2" extract was obtained with $343.52 \pm 3.09$ (mg Trolox/100 g leaves). The oral administration of verbascoside, a molecule identified in *Aloysia citriodora*, at a maximum dose of 3 mg/day as a dietary supplement in rabbits, demonstrated a protective effect on ocular tissue and fluids associated with oxidative stress. This protective effect was assessed by thiobarbituric acid reactive substances (TBARS) and the Trolox equivalent antioxidant capacity (TEAC) assay (*Mosca et al., 2013*). These results emphasize the necessity of future research on the extracts obtained in this study.

## Hydroxyl radical scavenging activity

The hydroxyl radical inhibition curve was determined using Oligopin (Nutri-Dyn, Maple Plain, MN, USA), which is recognized for its antioxidant power derived from natural pine bark extracts, as the standard. The results of this study are presented in Table 1. Oligopin, at a concentration of 1.04 mg/mL, demonstrated the most effective performance in terms of inhibition and the concentration utilized. For the aqueous extract, which achieved a lower concentration, T3R3A2 demonstrated the most effective result, with $0.72 \pm 0.007$ mg/mL inhibiting 50% of the radical.

In a study by *Valencia-Avilés et al. (2018)*, the inhibitory effect on the hydroxyl radical was investigated using oak extracts. The study found that the inhibition occurred at a concentration of $1,271 \pm 72$ µg/mL of Oligopin. In this study, the value obtained for Oligopin was $1.044 \pm 0.04$ mg/mL, which is similar to the values obtained for extract T3R3A2. According to the values obtained for the Trolox equivalent activity, there was higher activity at 75 °C and an amplitude of 35%, as well as a ratio of solvent to sample of 1:15. Research conducted by *Mehmood et al. (2024)* demonstrated that the methanolic extract of fresh *P. roxburghii* leaves exhibited the highest DPPH$^{\bullet}$ and OH radical scavenging activity, achieving 80% radical elimination with 200 µL of extract. Conversely, studies documented by *Olakanmi et al. (2024)* on the extraction with ethyl acetate from Croton *Albizia ferruginea* leaves yielded an IC$_{50}$ of 350 µg/mL for hydroxyl radical elimination, signifying elevated antioxidant activity in comparison to the present study, in which a value of 1.04 mg/mL was obtained. This difference could be attributed to the presence of lipophilic compounds, which may be present in the ethyl acetate extraction performed by *Olakanmi et al. (2024)*. This phenomenon has also been observed in studies reported by *Camarena-Tello et al. (2018)*. The researchers found an IC$_{50}$ in *Psidium guajava* leaf extracts of 1,103.42 µg/mL using chloroform. However, an IC$_{50}$ of 1,726.51 µg/mL was found using water for extraction, indicating that chloroform extraction is more efficient.

In addition to their demonstrated antioxidant value, the phenolic extracts of *Aloysia citriodora* show great potential for industrial applications. Their use in pharmaceutical

**Table 2** *p*-value levels from the three-way ANOVA for the effect of temperature (T), solute/solvent concentration (R), and amplitude (A).

|  | $T$ | $R$ | $A$ | $T \times R$ | $A \times R$ | $T \times A$ | $T \times R \times A$ | *RSquare* | *RMSE* | *Fratio* |
|---|---|---|---|---|---|---|---|---|---|---|
| Phenolic | * | * | * | * | * | * | * | 0.971 | 5.43 | 70.62 |
| Flavonoids | * | * | * | * | * | * | * | 0.994 | 3.00 | 355.7 |
| DPPH ($IC_{50}$) | * | * | * | * | * | * | * | 0.979 | 57.61 | 98.83 |
| ABTS ($IC_{50}$) | * | * | * | * | * | * | * | 0.997 | 1.02 | 892.9 |
| Hydroxyl ($IC_{50}$) | * | * | * | * | * | * | * | 0.996 | 0.14 | 648.8 |

Notes.

RMSE, Root mean square error; *T*, temperature; *R*, solute/solvent concentration; *A*, amplitude.

*Significant $p < 0.001$, $\alpha = 0.05$.

and nutraceutical formulations has been documented, owing to their capacity to modulate oxidative, inflammatory, and microbial processes (*Athanasiadis et al., 2024*). In the field of food science, these extracts have demonstrated prebiotic and antimicrobial properties, which are advantageous for the development of functional foods and natural preservatives. These extracts have been shown to stimulate the growth of beneficial bacteria, such as *Lactobacillus rhamnosus*, and to suppress common pathogens (*Gkalpinos et al., 2023*). From a technological standpoint, ultrasound-assisted extraction (UAE) is not only efficient and reproducible but also easily scalable to an industrial level, with lower solvent and energy consumption, making it a more environmentally sustainable option compared to traditional methods (*Hashemi et al., 2017*). In a study conducted by *Buchwald-Werner et al. (2018)*, the intake of 400 mg/day of *A. citriodora* extract (Recoverben®) led to a significant reduction in muscle strength loss in healthy, moderately active adults. These elements reinforce the applicability of the optimized extract in key sectors of the bioindustry. Although there are currently no studies of scale-up of ultrasound extraction using *Aloysia citriodora* leaves, numerous reports are available for other species relevant to food studies. Various studies support the scalability of UAE for phenolic compounds. *Rodríguez et al. (2022)* successfully scaled UAE to pilot scale (120 kg) for extracting polyphenols from olive pomace, achieving high yields (3.0 g GAE/L) under controlled conditions. *Panić et al. (2024)* demonstrated that UAE with natural deep eutectic solvents produces stable bioactive extracts suitable for functional foods and cosmetics. These findings support the industrial feasibility of applying UAE for obtaining *Aloysia citriodora* extracts.

## Multifactorial analysis

The following multifactorial analysis of variance (ANOVA) was conducted to determine whether there was a relationship between response variable $Y$ and certain categorical variables, which were the factors. The response variables included the phenolic compound content (mg GAE/g), flavonoid compound content (mg QE/g), and the $IC_{50}$ of DPPH•, ABTS•+, and hydroxyl radicals. The results of this study are shown in Table 2.

A three-way analysis of variance (ANOVA) was conducted on the data obtained from the $3^3$ experimental arrangements. A subsequent statistical analysis employing one-way analysis of variance (ANOVA) revealed discrepancies in the levels of phenolic compounds, flavonoid compounds, and the $IC_{50}$ values of hydroxyl, ABTS•+, and DPPH• radicals. The statistical significance of these findings was determined to be below the $\alpha = 0.05$ level of

significance, as shown in Table 2. Consequently, the null hypothesis was refuted, and the alternative hypothesis proposing a significant discrepancy in the extraction of flavonoids from *A. citriodora* by the ultrasound extraction method, contingent on temperature, solute/solvent ratio, and amplitude, was accepted.

As is shown in Table 2, an increase in temperature resulted in a corresponding increase in the extraction of total phenolic content (mg GAE/g). The temperature of 75 °C was found to yield the highest extraction yield. It was observed that a higher total flavonoid content (quercetin equivalents) was obtained at 50 °C, which corresponded to the ability to inhibit the hydroxyl radical at the same temperature. An estimated concentration of 1.65 mg/mL was required to eliminate 50% of the radical. In a similar manner, the other radicals exhibited superior $IC_{50}$ values at 50 °C. A correlation was identified between quercetin equivalents (QE) and the capacity to inhibit the radical. It was determined that an increase in the QE mg QE/g content of the extract was associated with a decrease in the amount of mg/mL of extract necessary to eliminate the hydroxyl radical. Several studies have demonstrated that extraction temperature exerts a substantial influence on the recovery of phenolic compounds. As reported by *Ozdemir et al. (2023)*, an increase in total phenolic content up to 70 °C was observed. This increase was attributed to greater cell disruption induced by ultrasonic cavitation. *Zampar et al. (2022)* examined the polyphenol content of compounds derived from pineapple byproducts. They found a maximum content of 405 mg GAE/100 g of compound at 60 °C. This result suggests that higher temperatures enhance the solubility and diffusion of phenolic compounds. In the study by *Oliveira et al. (2022)*, the researchers evaluated the effectiveness of ultrasound-assisted extraction and pressurized liquid extraction in *Citrus latifolia* Tan. It has been observed that elevated temperatures can enhance process efficiency, provided that the thermal degradation threshold is not exceeded. In the study, the highest yields of total phenolic compounds were obtained at 110 °C ($p < 0.05$), with a value of 17.66 mg GAE/g of dry weight. *Leliana et al. (2022)* have indicated that the optimal conditions for metabolite extraction from young coconut mesocarp using UAE entail a temperature of 70 °C and an extraction time of 5 min. In accordance with the data presented above, it is reaffirmed and emphasized that, for the present study, the optimal temperature for polyphenol extraction was 75 °C.

As illustrated in Fig. 2, the r values (least squares means) were derived from the multiple linear regression equation. The highest r values (total phenolic content of 92.70 mg GAE/g) were consistently obtained at a high temperature (75 °C), while the lowest (3.35 mg GAE/g) was obtained at a low temperature (30 °C). The contribution of the three factors (T, R, and A) was significant in yielding the highest r value (total phenolic content of 92.70 mg GAE/g). According to the findings of several studies, the optimal temperature range for extraction methods aimed at achieving the highest recovery of polyphenols is between 60 and 80 °C. In this study, it was observed that the highest antioxidant activity occurred within this temperature range (*Gironi & Piemonte, 2011*; *Antony & Farid, 2022*).

As shown in Table 2, the solute-to-solvent ratio exerts a significant influence on the extraction of phenolic and flavonoid compounds. The ratio of 1:10 (g/mL) yielded the highest phenolic content (gallic acid equivalents extraction), while a ratio of 1:15 yielded
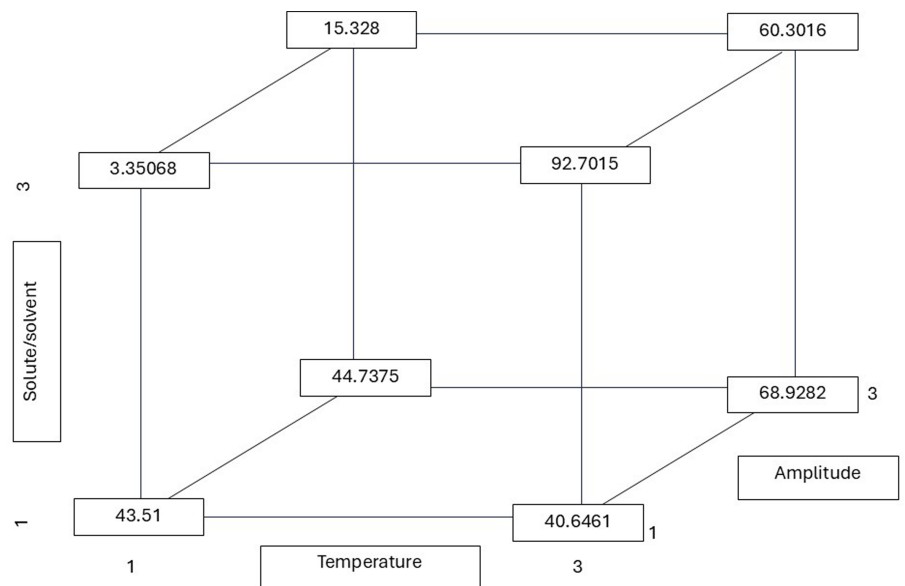

**Figure 2  Total phenolic r (least squares means) values calculated from the reported multiple linear regression equation considering all possible combinations of the temperature, solute/solvent concentration, and amplitude.** All possible combinations of three variables in the ultrasound-assisted extraction process were evaluated: temperature (30 °C, 50 °C, and 75 °C), solute/solvent ratio (1:5, 1:10, and 1:15), and ultrasound amplitude (20%, 25%, and 35%). The numerical values indicate the phenolic content (in mg of gallic acid equivalents per gram of dry extract) for each combination. This approach allows visualization of how the interaction between factors influences the release of phenolic compounds from the plant matrix. Notably, the highest phenolic content was obtained at 75 °C, 1:15, and 20% amplitude (92.7015 mg GAE/g), suggesting that higher temperatures may favor cell matrix disruption and the release of bound polyphenols.

a higher flavonoid content (quercetin equivalents). The 1:10 ratio resulted in a lower concentration, with the capacity to eliminate 50% of the radicals. The calculated r values from the multiple linear regression equation demonstrate the significance of the three factors (T, R, and A), yielding the highest r value (flavonoid content of 94.583 mg QE/g) (Fig. 3). This value was obtained at the highest temperature and ratio, with the amplitude having no significant influence and being at the lowest level to reach the highest r. *Dzah et al. (2020)* found that a low solvent–material ratio increased the dispersion and absorption of acoustic energy and attenuated the intensity of ultrasonic power, and *vice versa*. This effect influences the attraction of phenolic and antioxidant compounds. This phenomenon was observed in the present study, in which the extraction of phenolic compounds was found to be lower at higher organic matter concentrations. As stated by *Jovanović et al. (2021)*, the presence of a higher amount of extraction solvent (1:30 ratio) has been shown to prevent medium saturation and improve polyphenol content. In contrast, an excessive amount of plant material (at a ratio of 1:10) has been shown to increase viscosity, which in turn inhibits the expansion of ultrasonic waves and the diffusion of polyphenols *Jovanović et al. (2021)*. This finding is consistent with the individual analysis, as shown in Table 3,
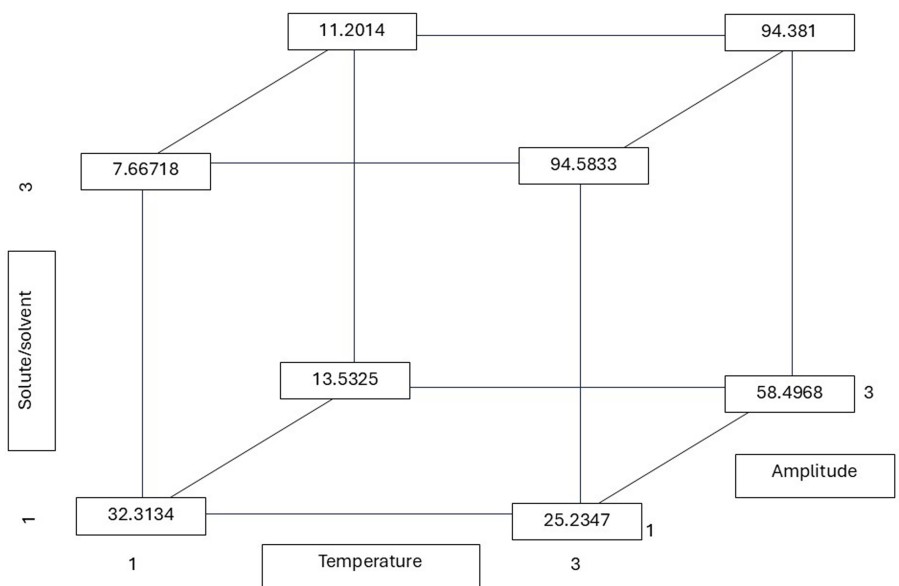

**Figure 3** **Total flavonoid r (least squares means) values calculated from the multiple linear regression equation considering all possible combinations of the temperature, solute/solvent concentration, and amplitude.** All possible combinations of three variables in the ultrasound-assisted extraction process were evaluated: temperature (30 °C, 50 °C, and 75 °C), solute/solvent ratio (1:5, 1:10, and 1:15), and ultrasound amplitude (20%, 25%, and 35%). The numerical values indicate the flavonoid content (in mg of quercetin equivalents per gram of dry extract) for each combination. This approach allows visualization of how the interaction between factors influences the release of phenolic compounds from the plant matrix. Notably, the highest flavonoid content was obtained at 75 °C, with a 1:15 ratio and 20% amplitude (94.58 mg QE/g), suggesting that higher temperatures may favor the disruption of the cell matrix and the release of bound flavonoids.

which demonstrates a significant difference in higher flavonoid content at a 1:10 ratio compared to the other dilutions examined in this study.

Table 2 shows a positive correlation exists between the amplitude and the extraction of total phenolic and total flavonoid compounds. The highest amplitude (35%) yielded the highest flavonoid content (equivalent to quercetin extraction). Regarding radicals, there were no differences between amplitudes of 25 and 35%; differences were observed only between the low amplitude (20%), which resulted in a higher concentration to eliminate 50% of the radicals. In a recent study, *Babotă et al. (2022)* reported an ultrasound amplitude of 34.8% for the extraction of phenolic compounds from *Thymus comosus* Heuff. In the present study, the optimal amplitude for the extraction of individual flavonoids was determined to be 35%. In a series of studies conducted within our research group (*Tena-Rojas et al., 2022*; *Tranquilino-Rodríguez & Martínez-Flores, 2023*; *Garnica-Romo, Sanchez-Pahua & Martinez-Flores, 2024*), the extraction of phenolic compounds from *P. guajava*, *Moringa oleifera*, and *A. citriodora* leaves was examined. This extraction was performed using an amplitude ranging from 30% to 50% during ultrasound-assisted extraction. The findings of these studies indicated optimal extraction yields of total phenolic compounds, total flavonoid compounds, and antioxidant capacity.

**Table 3  Effect of temperature, solute/solvent ratio, and amplitude on response variables during the extraction process.**

| | Total phenolic content (mg GAE/g) | Total flavonoid content (mg QE/g) | $OH^-IC_{50}$ (mg/mL) | $DPPH^\bullet IC_{50}$ ($\mu$g/mL) | $ABTS^{\bullet+}IC_{50}$ ($\mu$g/mL) |
|---|---|---|---|---|---|
| Temperature | | | | | |
| 30 °C | 32.86[C] | 23.28[C] | 3.32[A] | 425.95[A] | 42.58[A] |
| 50 °C | 60.37[B] | 71.05[A] | 1.65[C] | 130.65[B] | 22.59[C] |
| 75 °C | 68.99[A] | 61.01[B] | 1.98[B] | 107.95[B] | 24.46[B] |
| Solute/solvent Ratio | | | | | |
| 1/5 | 50.22[B] | 44.01[C] | 2.43[B] | 175.85[B] | 38.40[A] |
| 1/10 | 58.61[A] | 47.22[B] | 1.75[C] | 163.78[B] | 24.80[C] |
| 1/15 | 53.37[B] | 64.12[A] | 2.79[A] | 324.93[A] | 26.43[B] |
| Amplitude | | | | | |
| 20 | 48.28[C] | 48.23[B] | 3.25[A] | 362.17[A] | 38.23[A] |
| 25 | 60.29[A] | 47.31[B] | 1.69[C] | 151.36[B] | 25.44[B] |
| 35 | 53.64[B] | 59.80[A] | 2.01[B] | 151.02[B] | 25.97[B] |

**Notes.**
Different letters in the same column for each parameter (temperature, solute/solvent ratio, and amplitude) indicate significant differences between treatments ($p < 0.05$).

As illustrated in Fig. 4, the calculated r values from the multiple linear regression equation demonstrate that the contribution of the three factors (T, R, and A) was significant in obtaining a low r value (0.86 mg/mL). The value for the hydroxyl radical was obtained under conditions of elevated temperature and solute-to-solvent ratio. The amplitude also had an influence, being at the lowest level to obtain the r value. The graphs depicting $DPPH^\bullet$ and $ABTS^{\bullet+}$ are not presented due to their lack of statistical significance in comparison to the hydroxyl radical data.

Table 3 presents the root mean square error (RMSE) and the coefficient of determination ($R^2$). The model's predictions regarding the hydroxyl radical and the phenolic content demonstrated a satisfactory degree of accuracy, as evidenced by the high degree of fit observed. The $F$-ratio values were found to be elevated, as evidenced by the high values observed in the case of $ABTS^{\bullet+}$ (648.8), suggesting that the model exerted a significant influence on this particular response variable. The findings of the multifactorial analysis indicated that treatment T3R3A2, which corresponded to a temperature of 75 °C, an amplitude of 25%, and a solute/solvent ratio of 1:15, exhibited the optimal capacity to inhibit hydroxyl and ABTS radicals. Furthermore, the study demonstrated the highest total phenolic compound content and a significant flavonoid compound content, suggesting its potential as a promising therapeutic agent. In order to enhance the statistical interpretation and facilitate a more profound comprehension of the impact of independent variables (temperature, solute/solvent ratio, and ultrasonic amplitude), readers are advised to refer to the Supplemental Information, in which the prediction equations and fitting models for each response variable are provided, including flavonoid content, total polyphenol content, and free radical scavenging capacity (ABTS, DPPH, and hydroxyl radical).

Several studies have demonstrated the efficacy of green extraction technologies, such as microwave-assisted extraction (MAE), ultrasound-assisted extraction (UAE), and

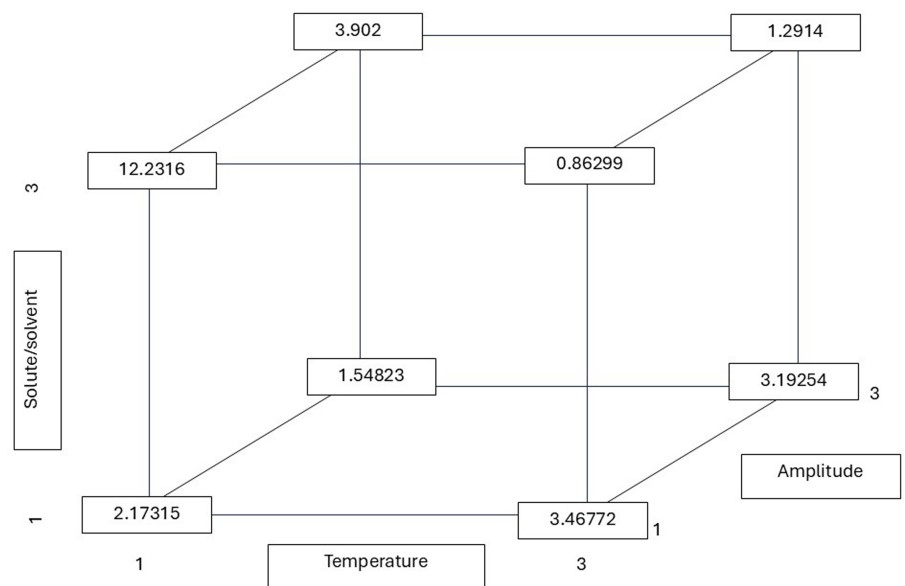

**Figure 4 Hydroxyl radical r (least squares mean) values calculated from the multiple linear regression equation considering possible combinations of the temperature, solute/solvent concentration, and amplitude.** All possible combinations of three variables in the ultrasound-assisted extraction process were evaluated for their effect on hydroxyl radical scavenging activity ($IC_{50}$ values): temperature (30 °C, 50 °C, and 75 °C), solute/solvent ratio (1:5, 1:10, and 1:15), and ultrasound amplitude (20%, 25%, and 35%). The numerical values in the diagram represent the least squares means of $IC_{50}$ (in mg/mL) for each factor level combination, calculated using a multiple linear regression model. This approach helps to visualize how the interaction of extraction parameters influences antioxidant potential. Notably, the lowest $IC_{50}$ value—indicating the highest hydroxyl radical scavenging activity—was observed at 75 °C, a 1:15 solute/-solvent ratio, and 25% amplitude (0.86299 mg/mL), suggesting that this combination enhances the release of antioxidant compounds capable of neutralizing hydroxyl radicals.

supercritical fluid extraction (SFE), in the recovery of phenolic compounds from plant matrices, including aromatic species such as *Aloysia citriodora*. These techniques have been shown to enhance extraction yield and reduce processing times, while also reducing or eliminating the use of toxic solvents. This aligns with principles of sustainability (*Ameer, Shahbaz & Kwon, 2017*). In this context, *Leyva-Jiménez et al. (2019)* evaluated the use of microwave-assisted extraction in *Aloysia citriodora* leaves, observing that the extraction time influenced the recovery of phenylpropanoids, with shorter times resulting in higher efficiency. In contrast, studies by *Garnica-Romo, Sanchez-Pahua & Martinez-Flores (2024)* on *Aloysia citriodora* leaves identified a robust correlation between the $IC_{50}$ values for DPPH• and ABTS•+ radical scavenging activity and the total phenolic content in the extracts. These studies also confirmed that ultrasound-assisted extraction produced higher yields in less time compared to maceration. Furthermore, *Parodi et al. (2013)* identified the presence of cinerolone, a type of ketoalcohol, in *Aloysia citriodora* leaf extracts obtained through supercritical fluid extraction. Finally, *Athanasiadis et al. (2024)* demonstrated that optimizing the extraction process of *A. citriodora* using 50% ethanol, high temperatures (80 °C), and technologies such as ultrasonication and pulsed electric fields (PEF) significantly enhances the recovery of bioactive compounds. These extracts,

which are abundant in total phenols and antioxidants, demonstrate considerable promise for utilization in pharmaceutical and food applications. Additionally, the equations used to estimate the values shown in Figs. 2–4 are included in the Supplementary Material, along with contour plots that visualize the interactions between factors such as temperature, solute/solvent ratio, and amplitude.

## CONCLUSIONS

The multifactorial analysis, based on the optimization of ultrasound processing conditions, allowed for the identification of the most effective extraction conditions for compounds from *A. citriodora* Palau leaves. The analysis revealed that the greatest effect was achieved at a temperature of 75 °C, a wave amplitude of 25%, and a solute to solvent ratio of 1:15. These conditions resulted in the highest amount of phenolic compounds extracted, reaching 87.74 mg GAE/g of sample, along with a notable capacity to inhibit hydroxyl and 2,2′-azino-bis(3-ethylbenzothiazoline-6-sulfonic acid) (ABTS$^{\bullet+}$) radicals, with values of 0.72 mg/mL and 9.08 µg/mL, respectively, in addition to a flavonoid content of 76.82 mg QE/g. Consequently, this treatment emerges as a promising strategy for obtaining extracts with high antioxidant capacity and therapeutic potential. Subsequent research should concentrate on optimizing other process variables, such as ultrasound duration or the utilization of different solvents, with the objective of further enhancing extraction yield and the biological activity of the compounds obtained. It is further recommended that additional *in vivo* studies be conducted to validate and explore the therapeutic potential of the extract, as well as to identify the metabolites present in the optimal extract using techniques such as HPLC. These steps are expected to facilitate the consolidation of the applicability of the obtained extracts in the treatment of various diseases and to expand their pharmacological potential. This study emphasizes the significance of process optimization in the development of bioactive extracts, which establishes the foundation for their integration into therapeutic applications. It is important to note that results obtained from *in vitro* antioxidant assays are not always directly extrapolable to *in vivo* conditions. This is due to factors such as bioavailability, metabolism, and distribution of compounds within the organism. Consequently, experimental validation in biological models is required to confirm their actual effectiveness. From an industrial and pharmaceutical perspective, ultrasound-assisted extraction offers a clean and efficient alternative for obtaining natural bioactive compounds. *Aloysia citriodora* extracts, characterized by their high antioxidant activity, have the potential to serve as functional ingredients in a variety of products, including foods, cosmetics, and pharmaceuticals. This study provides an initial basis for the development of natural products and promotes the use of more sustainable extraction methods.

### Funding

The authors received no funding for this work.

## Competing Interests

The authors declare there are no competing interests.

## Author Contributions

- Osvaldo Alvarez Cortes conceived and designed the experiments, performed the experiments, analyzed the data, prepared figures and/or tables, authored or reviewed drafts of the article, and approved the final draft.
- Héctor Eduardo Martinez Flores conceived and designed the experiments, performed the experiments, analyzed the data, prepared figures and/or tables, authored or reviewed drafts of the article, and approved the final draft.
- Martha-Estrella García-Pérez conceived and designed the experiments, analyzed the data, authored or reviewed drafts of the article, and approved the final draft.
- Juan Carlos González-Hernández conceived and designed the experiments, analyzed the data, authored or reviewed drafts of the article, and approved the final draft.
- Martha Viveros conceived and designed the experiments, analyzed the data, authored or reviewed drafts of the article, and approved the final draft.
- Mariano Martinez-Vazquez conceived and designed the experiments, analyzed the data, authored or reviewed drafts of the article, and approved the final draft.
- Noemi Silva Jimenez conceived and designed the experiments, analyzed the data, authored or reviewed drafts of the article, and approved the final draft.
- Maria Carmen Bartolome Camacho conceived and designed the experiments, analyzed the data, authored or reviewed drafts of the article, and approved the final draft.

## Data Availability

   The raw measurements are available in the Supplementary File.

## Supplemental Information

Supplemental information for this article can be found online at http://dx.doi.org/10.7717/peerj.19821#supplemental-information.

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
