# Peer review of "Multifactorial analysis of temperature, solute-to-solvent ratio, and ultrasound amplitude on the extraction of phenolic and antioxidant compounds from Aloysia citriodora Palau leaves"

_PeerJ, doi:10.7717/peerj.19821_

## Round 0.1 · original submission · Major Revisions

·

Basic reporting

Strengths:
1. The article is generally well-written with appropriate scientific language.
2. Background information is sufficient and includes up-to-date literature.
3. Figures and tables are relevant, clearly labeled, and well-referenced in the text.
4. Keywords and abstract are informative and match the manuscript content.
5. The study adheres to PeerJ’s structural and formatting guidelines.
Suggestions for Improvement:
1. There are occasional grammar and phrasing issues, e.g., line 31 ("An ultrasonic assisted extraction was applied...") should be revised to "Ultrasonic-assisted extraction was applied....
2. A language edit could further improve clarity and readability.
3. Some figures are mentioned without visual inserts ("Insert Figure 1", etc.). Ensure all are present and correctly placed.

Experimental design

Strengths:
1. The use of a full factorial 3³ design is appropriate for evaluating multiple interacting factors.
2. The methodology for antioxidant capacity assessment is robust and uses multiple radical scavenging assays (ABTS, DPPH, OH).
3. Statistical treatment (ANOVA, regression analysis) is comprehensive and supports the conclusions.
Suggestions:
1. While the methods are described in detail, including more information on how extract stability was controlled during and after processing would be helpful.
2. The ultrasound device settings (frequency, probe type) should be specified for reproducibility.
3. Justify why a 10-minute extraction time was chosen when other studies mentioned longer durations.

Validity of the findings

Strengths:
1. The conclusions align with the results obtained and do not overreach.
2. Control compounds (Trolox, gallic acid, ascorbic acid) strengthen the findings’ credibility.
3. IC50 values are well-documented, and the treatment T3R3A2 is convincingly shown as optimal.
Suggestions:
1. Discuss potential variability due to plant age, harvest season, or drying methods in the context of extract efficacy.
2. Mention limitations regarding extrapolation of in vitro antioxidant assays to in vivo applications.

Additional comments

Overall Evaluation: This is a scientifically sound manuscript with significant value in optimizing plant-based antioxidant extraction techniques. The results are of potential interest to researchers in food science, pharmacognosy, and natural product chemistry.
Specific Suggestions:
1. Add a schematic of the experimental workflow to enhance understanding of the factorial design.
2. Consider comparing ultrasound extraction with other modern techniques briefly (e.g., microwave or pressurized liquid extraction) in the discussion.
3. Expand the conclusion to include a brief discussion of industrial or pharmaceutical implications.

·

Basic reporting

The manuscript is generally well-structured, written in clear and professional English, and adheres to PeerJ’s formatting guidelines. The introduction effectively establishes the study's relevance and contextualizes it within current research on ultrasonic-assisted extraction of bioactive compounds. The authors provide an adequate review of the literature, and references are both current and relevant.

Figures and tables are relevant to the data and conclusions. However, figure captions (especially Figures 2–4) would benefit from further elaboration to aid standalone interpretation. Raw data and supplementary materials appear to be complete and accessible.

Experimental design

The experimental design is appropriate and scientifically sound. The use of a 3³ full factorial design is well-justified and enables a thorough exploration of parameter effects and interactions. The selection of ultrasound-assisted extraction, along with phenolic and flavonoid content and three antioxidant assays (DPPH, ABTS, OH), provides a comprehensive evaluation.

The description of materials and methods is sufficiently detailed to allow replication. Nonetheless, the rationale for fixing extraction time at 10 minutes, although mentioned briefly, could be reinforced by citing specific pilot data or comparative studies.

Validity of the findings

To strengthen the discussion, the authors should more clearly acknowledge the limitations of their work—for example, the use of only water as the extraction solvent (excluding comparisons with organic solvents), the lack of compound-specific identification (e.g., HPLC or LC-MS profiling), and the absence of in vivo testing to confirm biological effects.

Additional comments

- This study represents a valuable contribution to the optimization of extraction parameters for A. citriodora using green processing methods.

- Clarify figure captions to enhance clarity and reduce the need to cross-reference with the text.

- Ensure consistency in the use of units (e.g., µg/mL, mg/mL) and spacing across all tables and figures.

- Consider adding a brief section discussing the potential industrial applications of the optimized extract (e.g., nutraceutical, pharmaceutical, or functional food formulations).

- A brief mention of scalability and environmental aspects of the ultrasonic process would enhance the broader applicability of the findings.

Reviewer 3 ·

Basic reporting

The introduction lacks a strong scientific rationale and fails to clearly define the motivation behind the study. The advantages of optimizing the solute/solvent ratio, ultrasound amplitude, and temperature for the extraction of phenolic compounds are not sufficiently explained. The authors should better contextualize the significance of such optimization, highlighting potential benefits for industrial, environmental, or functional applications.

In section 2.1 (line 84), the botanical maturity of the A. citriodora leaves and the criteria used for collection must be described. If such information is detailed in section 3.1, this should be explicitly referenced.

Lines 58–59: The reference to Cannabis is not relevant in a study focused on Aloysia citriodora. This example should be revised or replaced with a more appropriate species.

Experimental design

Lines 95–98: While the rationale for the selected ultrasound amplitude is provided, the same is not true for the chosen solute/solvent ratios. The authors should clarify whether these values were based on previous literature, preliminary tests, or equipment constraints.

The ultrasound amplitude should be expressed in Watts rather than percentage. Additionally, the complete manufacturer information for the VCX 500 device should be included.

The calibration curves used for total phenolic content (TPC) and total flavonoid content (TFC) determination should be provided as supplementary material.

The equations used for the calculation of DPPH inhibition percentage, TEAC, and hydroxyl radical scavenging activity should be clearly presented within the manuscript.

The units used to express antioxidant activity results must be standardized throughout the manuscript (e.g., µmol TE/g, IC₅₀, % inhibition, etc.).

Several instances (lines 228, 273, 284, among others) present the IC₅₀ notation incorrectly. The subscript formatting must be corrected across the entire manuscript.

Validity of the findings

Table 1 lacks central point replications, which compromises the ability to assess the reproducibility of the data. Moreover, no independent extraction was performed to confirm the optimal extraction condition.

Given that a full factorial design was employed, the authors could have expanded the study with a response surface methodology (RSM) to better explore and optimize the extraction conditions. This would add significant scientific value to the manuscript.

Table 3 reports an unexpectedly higher TPC at 75 °C compared to 30 °C and 50 °C. Since phenolic compounds are generally heat-sensitive, this result needs to be thoroughly explained. The authors should discuss possible reasons, such as enhanced matrix disruption or release of bound phenolics at higher temperatures.

Additional comments

The manuscript addresses a relevant topic and presents potentially interesting experimental data. However, major revisions are needed in the introduction, methodological descriptions, and data interpretation. Once these issues are properly addressed, the manuscript will be significantly strengthened and better positioned to contribute to the field of green extraction of bioactive compounds.

---

## Round 0.2 · accepted · Accept

The reviewers have concluded that all requested revisions were completed, and that the manuscript is now suitable for publication. Congratulations.

·

Basic reporting

The authors address all necessary comments raised.

Experimental design

-

Validity of the findings

-

·

Basic reporting

The manuscript is well-structured and generally written in clear, professional English. The abstract effectively summarizes the study’s scope and main findings. The introduction provides appropriate context and a relevant literature review, particularly highlighting the significance of ultrasound-assisted extraction and its application to Aloysia citriodora.

Figures and tables are relevant and support the text. In the revised version, the figure captions (especially Figures 2–4) have been sufficiently improved for clarity and self-interpretation, as previously suggested.

References are current, appropriately cited, and relevant to the topic. Supplementary materials appear complete and well-integrated into the manuscript.

Overall, the basic reporting criteria have been fully met.

Experimental design

The experimental design is appropriate, rigorous, and well-suited to the study objectives. The use of a full 3³ factorial design allows for a comprehensive assessment of the interactive effects of temperature, solute-to-solvent ratio, and ultrasound amplitude on the extraction of phenolic and flavonoid compounds, as well as antioxidant activity.

The revised manuscript provides sufficient methodological detail to ensure reproducibility. Notably, the authors have responded well to previous comments by justifying the fixed extraction time of 10 minutes with supportive references and preliminary data. Additionally, the use of three antioxidant assays (DPPH, ABTS, OH•) strengthens the experimental depth and reliability of the findings.

Overall, the experimental procedures are clearly described, scientifically sound, and support the study’s conclusions.

Validity of the findings

The findings are valid, clearly presented, and supported by appropriate statistical analyses. The revised manuscript effectively integrates a multifactorial ANOVA and presents comprehensive data on total phenolic and flavonoid contents, as well as antioxidant activity across multiple treatments. The inclusion of IC50 values for ABTS, DPPH, and hydroxyl radicals enhances the credibility of the results.

The authors have also acknowledged key limitations of their study, particularly the exclusive use of water as the extraction solvent, the absence of compound-specific identification (e.g., via HPLC), and the lack of in vivo validation. These points are appropriately addressed in the revised conclusion and discussion, outlining future directions for research and application.

Together, the experimental outcomes, statistical support, and transparent discussion of limitations indicate that the findings are robust, reproducible, and valid within the scope of the study.

Additional comments

This study represents a valuable contribution to the field of green extraction technologies and bioactive compound recovery from Aloysia citriodora using ultrasound-assisted extraction (UAE). The authors have carefully addressed all previous reviewer comments, and the revised manuscript shows substantial improvement in clarity, detail, and applicability. Overall, the manuscript is now well-structured, clear, and ready for publication.

Reviewer 3 ·

Basic reporting

From my perspective, all reviewer comments have been properly addressed, and the manuscript is ready for publication.

Experimental design

-

Validity of the findings

-